# Ochratoxin A in Slaughtered Pigs and Pork Products

**DOI:** 10.3390/toxins14020067

**Published:** 2022-01-19

**Authors:** Mikela Vlachou, Andreana Pexara, Nikolaos Solomakos, Alexander Govaris

**Affiliations:** Laboratory of Hygiene of Foods of Animal Origin, Faculty of Veterinary Science, University of Thessaly, 43100 Karditsa, Greece; vlachoumikvet@uth.gr (M.V.); nsolom@vet.uth.gr (N.S.); agovaris@vet.uth.gr (A.G.)

**Keywords:** ochratoxin A, mycotoxins, slaughtered pigs, pork products

## Abstract

Ochratoxin A (OTA) is a mycotoxin that is produced after the growth of several *Aspergillus* and *Penicillium* spp. in feeds or foods. OTA has been proved to possess nephrotoxic, hepatotoxic, teratogenic, neurotoxic, genotoxic, carcinogenic and immunotoxic effects in animals and humans. OTA has been classified as possibly carcinogenic to humans (Group 2B) by the IARC in 2016. OTA can be mainly found in animals as a result of indirect transmission from naturally contaminated feed. OTA found in feed can also contaminate pigs and produced pork products. Additionally, the presence of OTA in pork meat products could be derived from the direct growth of OTA-producing fungi or the addition of contaminated materials such as contaminated spices. Studies accomplished in various countries have revealed that pork meat and pork meat products are important sources of chronic dietary exposure to OTA in humans. Various levels of OTA have been found in pork meat from slaughtered pigs in many countries, while OTA levels were particularly high in the blood serum and kidneys of pigs. Pork products made from pig blood or organs such as the kidney or liver have been often found to becontaminated with OTA. The European Union (EU) has established maximum levels (ML) for OTA in a variety of foods since 2006, but not for meat or pork products. However, the establishement of an ML for OTA in pork meat and meat by-products is necessary to protect human health.

## 1. Introduction

Ochratoxin A (OTA) is a mycotoxin that is produced by several fungal species of the genera *Aspergillus* and *Penicillium* in a wide variety of agricultural commodities during the field period or storage worldwide. OTA was found to possess a carcinogenic effect in animals and poultry, in addition to its nephrotoxic, hepatotoxic, teratogenic, neurotoxic, genotoxic and immunotoxic effects in animals [1,2,3,4,5,6]. OTA has been also classified as possibly carcinogenic to humans (group 2B) by the International Agency for Research on Cancer [7]. Thus, the European Commission (EC) has established maximum levels (ML) for OTA in these commodities [8].

OTA found in consumed feedstuffs [9] can adversely affect animal health and decrease the production (e.g., milk) of animals [10]. The same mycotoxin could also increase susceptibility to secondary bacterial infections in growing pigs, and immunosuppression is the first expressed toxic effect of OTA [11]. Ruminants are less sensitive to OTA intoxication as compared to monogastric animals. In ruminants, microflora in rumen can degrade OTA to the virtually non-toxic ochratoxin α (OTα) [12]. Moreover, the exposure of food-producing animals to OTA via feed consumption can result in undesirable OTA residues in animal-derived food products (“carry-over effect” in meat, eggs or milk), contributing to the human intake of OTA via indirect transmission [13].

Pigs are the most susceptible animals to OTA exposure as compared to other productive animals [14,15]. High OTA occurrence has been recorded in feed ingredients and finished swine feeds in various countries [16,17,18,19], highlighting the pig exposure to the toxin. OTA can accumulate in several pig tissues, with the highest concentrations found in blood, followed by the kidneys and liver, whereas lower concentrations have been found in muscles and fat [20,21,22,23].

The contamination of pork meat/edible offal with OTA is mainly derived from the consumption of OTA-contaminated feed by the pigs [24,25]. Additionally, the presence of OTA in pork meat products (ham muscle, cured meats, salami or dry-cured ham) could be derived from the direct growth of OTA-producing fungi, such as *Penicillium nordicum* and *Penicillium verrucosum* [26,27,28,29,30], or from the addition of OTA-contaminated materials such as contaminated spices [31,32].

Pork meat and meat products are amongst the important sources of chronic dietary exposure to OTA in humans [14]. Nevertheless, the EU has not established maximum OTA levels in pork meat and pork-derived products. Some European countries have adopted regulations or guidelines on OTA concentrations in meat and/or meat products, such as Denmark, Estonia, Romania, Slovakia and Italy [33]. No binding limits on OTA in meat and meat products have been set in the USA, Australia, Canada and Asia [34,35].

Several studies have been conducted worldwide on the occurrence of OTA in pork meat and pork-derived products. This review summarizes the state-of-the-art on OTA and focuses on the occurrence of OTA in slaughtered pigs and pork products.

## 2. Production of OTA

### 2.1. General Factors

OTA is mainly produced by *Aspergillus* and *Penicillium* fungi species in a wide variety of agricultural commodities, livestock products and processed food [36]. The majority of OTA-producing *Aspergillus* species belong to *Circumdati* section (*A. ochraceus* group, *A. steynii*, *A. westerdijkiae*) and *Nigri* section (*A. carbonarius* and *A. niger*) [37,38].

Apart from the main OTA toxigenic species *A. ochraceus* and *A. westerdijkiae*, several other *Aspergillus* species have been linked to OTA production, such as *A. sclerotiorum*, *A. sulphureus*, *A. albertensis*, *A. auricomus*, *A. wentii*, *A. fumigates*, *A. versicolor*, *A. cretensis*, *A. flocculosus*, *A. pseudoelegans*, *A. roseoglobulosus*, *A. sulphurous* [39], *A. alliaceus* [40], *A. welwitschiae* (formerly *A. awamori*) [41], *A. affinis* [42], *A. steynii* [43], *A. lacticoffeatus*, *A. sclerotioniger* [44] and *A. carbonarius* [45].

The OTA toxigenic *Penicillium* genus is classified in groups of *P. verrucosum* and *P. nordicum* [46] or *P. thymicola* [47]. *Penicillia* species of *P. chrysogenum*, *P. glycyrrhizacola*, *P. polonicum* [48], *P. brevicompactum*, *P. crustosum*, *P. olsonii*, *P. oxalicum* [49], *P. nalgiovense*, *P. solitum*, *P. salamii* [50] and *P. commune* [51] have been reported to produce OTA.

Important factors of fungal OTA production are considered to be temperature, water activity (a_w_) and growth medium composition. *Aspergillus* species predominate in warm climate regions, while *Penicillium* isolates are frequently found in cold climate regions [39]. *A. ochraceus* is commonly found in hot-tropical and semitropical climates with temperatures ranging between 12 to 37 °C and an a_w_ of growth substrates up to 0.77 [52,53]. The optimum temperature of OTA production by *A. ochraceus* is in the range of 25 and 30 °C, and the optimum a_w_ is 0.98 [39]. In tropical regions, toxigenic OTA *A. westerdijkiae* strains are frequently found [43]. In Europe and Canada, *P. verrucosum* is mostly associated with OTA production in cereals, with optimum growth temperatures of 4 to 31 °C and an optimum a_w_ up to 0.8 [54,55]. *P. viridicatum* can produce OTA at wider ranges of temperatures from 4 to 30 °C [56]. *P. nordicum* can grow on substrates with high protein and salt (5% NaCl) content [57], with the maximum growth observed at 20 °C, while the highest OTA production was found at 25 °C [24].

The growth of *Aspergillus* and *Penicillium* fungi species in feedstuffs is a significant issue for an OTA-safe feed chain supply. OTA-contaminated cereal and cereal by-products, which are important ingredients in pig feeds, are fueling concerns over the contamination with OTA of produced pork meat [9,58]. Several studies have indicated the contamination of pigs feeds with OTA [16,17,18]. The EC has established guidance values for OTA concerning complementary and complete feeding stuff, recommending a maximum concentration of 50 μg/kg for pigs [59,60]. In Brasil, Rosa et al. [19] reported that corn samples (44%) were contaminated with 42–224 μg/kg of OTA, while swine feed (31%) was contaminated with 36–120 μg/kg of OTA, respectively. In Costa Rica, 19 out of 57 feed samples were found to contain more than 50 mg/kg of OTA in pigs and sows [12]. In Italy, Pozzo et al. [17] found swine feed samples contaminated with OTA at 0.22–38.4 μg/kg levels. In the Czech Republic, Zachariasova et al. [61] found 26 samples of pig feeds contaminated with a mean OTA concentration of 3 μg/kg, with only one sample exceeding the EU recommended OTA value of 50 μg/kg. In China, Li et al. [18] found that 3% of the examined complete swine feeds exceeded Chinese regulatory limits for feedstuffs, set at 100 μg/kg for OTA.

### 2.2. Production of OTA in Meat and Meat Products

The most important *Aspergillus* and *Penicillium* OTA-producing species in meat and meat products are presented in Table 1. OTA toxigenic *Aspergillus* and *Penicillium* genera can also grow on the surface of dry cured-meat products due to their tolerance to low pH and high salt concentration [31,57]. The formation of OTA in the outer case or inner part of dry-cured meat products by *Aspergillus* or *Penicillium* was also examined [62,63]. The OTA toxigenic *P. nordicum* and *P. verrucosum* have been isolated in dry meat products in several European countries [64,65,66,67,68,69]. *P. nordicum* is a psychrotrophic fungi and widely distributed contaminant in various meat products [70,71,72,73,74]. It is considered an important OTA producer in dry-cured ham and dry-cured fermented sausage [24,28,70,75,76,77,78]. The ability of *P. nalgiovense*, *P. chrysogenum*, *P. olsonii*, *P. solitum* and *P. salamii* to produce OTA in dry meat products has been reported in previous studies [50,79].

*Aspergillus* spp. are less common OTA toxigenic contaminants than *Penicillium* spp. in dry-cured meats, since these products are processed at a low temperature and *Aspergillus* spp. usually requires higher temperatures than *Penicillium* spp. [80]. The occurrence of *A. westerdijkiae* has already been described in various meat products in many countries worldwide [30,81,82,83], and its growth was associated with the presence of the OTA toxin in dry-cured meat products [84,85,86]. *A. ochraceus* was detected in OTA-contaminated sausages [75] or dry cured ham [27,76].

In recent studies, the relationship between OTA production and the expression of the genes potentially involved in OTA biosynthesis have been examined in meat products [87,88,89].

**Table 1 toxins-14-00067-t001:** OTA producing *Aspergillus* spp. and *Penicillium* spp. in meat and meat products.

Species	Foodstuffs	References
*Aspergillus* spp.		
*A. ochraceus*	Processed meat	[90]
	Meat	[91]
	Dry-cured ham	[27]
	Sausages, speck	[75]
	Dry-cured meat products	[57]
*A. niger*	Dry-cured meat products	[57]
*A. westerdijkiae*	Meat products	[30,70,84,85,92,93]
*Penicillium* spp.		
*P. verrucosum*	Sausage casings	[94]
	Sausages	[75]
	Dry-cured meat products	[57]
*P. nalgiovense*	Dry cured ham	[95]
	Dry-cured meat (salami)	[50]
*P. nordicum*	Meat products	[46,96]
	Dry-cured ham	[64,65,95]
	Meat	[97]
	Cured meat	[98]
	Fermented meat	[99]
	Fermented sausage, liver pâté	[100]
	Dry-cured meat	[70,71,84]
	Sausage casings	[94]
	Sausages	[75,101]
	Meat product (Italian culatello)	[93]
	Salami	[28,67]
	Speck	[76]
*P. salami*, *P. solitum*, *P. chrysogenum*, *P. olsonii*	Dry-cured meat (salami)	[50]

## 3. Physicochemical Properties of OTA

The chemical name of OTA is L-phenylalanine-N-[(5-chloro-3,4-dihydro-8-hydroxy-3-methyl-1-oxo-1H-2-benzopyrane-7-yl)carbonyl]-(R)-isocoumarin. OTA consists of an isocumarin nucleus bonded to an L-phenylalanine unit by an amide bond [3].

After OTA ingestion in human or animals, several OTA derivatives are transformed. Certain OTA derivatives are hydroxylated, while others lack phenylalanine moiety or are conjugated (e.g., with glutathione, glucuronic acid, sulfate or pentose). Among the OTA metabolites are the dechloro analog ochratoxin B (OTB), the dechloro analog ochratoxin β (OTβ), the ethyl ester ochratoxin C (OTC) and the isocoumaric derivative OTα [32]. The OTA metabolites usually possess low or no toxicity [102,103].

OTA is stable to heat treatments and low pH conditions. Typical heat treatments of foods such as boiling, baking, frying and roasting do not cause any important decrease in OTA levels [104,105]. Pleadin et al. [106] found that the 30 min cooking (100 °C) and frying (170 °C) of contaminated sausages proved insufficient to decrease OTA levels.

Innovative food-processing methods such as irradiation, cold plasma (CP) and high-pressure processing (HPP) may be an effective means of OTA decrease in foods by controlling the growth of OTA-producing fungi [107,108]. Despite the promising results of these new technologies in the treatment of OTA-contaminated food, there is a great concern regarding the toxicity of the OTA degradation compounds and consequent implications on human and animal health [109]. The majority of innovative food-processing studies have been conducted in food of plant origin and not in foods of animal origin.

## 4. Toxicity of OTA

Several studies have indicated that OTA has nephrotoxic, hepatotoxic, teratogenic, neurotoxic, genotoxic and immunotoxic effects and can cause tumors in organs and tissues such as the kidneys, liver, intestine, ureters, lung, oculi and muscles of animals and humans [5,6,35,60,110]. The consumption of OTA-contaminated feed can adversely affect animals’ health and has been associated with several animal diseases, including porcine nephropathy, avian ochratoxicosis and carcinogenicity in rodents and poultry [5,6,60,111].

In humans, dietary exposure to OTA represents a serious health issue, including endemic nephropathies and urinary tract tumors [112]. OTA has been classified as possibly carcinogenic to humans (Group 2B) by the IARC due to evidence of OTA-mediated carcinogenicity in laboratory animals [7].

In many animal species and in humans, the primary target organ of OTA is the kidney [14]. The nephrotoxic effect of the toxin has been demonstrated by various studies performed on laboratory animals [2,35]. Severe nephrotoxicity has been detected in rats [113], with renal damages characterized by disorganization of the tubules, apoptosis, polyploidy in the proximal convoluted tubule (PCT) and an increase in the nucleus–cytoplasm ratio of their kidneys [114,115,116,117]. Furthermore, tubular nephrosis and hemorrhage were detected in rat kidneys [118]. Similarly, hemorrhage, tubular necrosis, mesangial hypercellularity and glomerulosclerosis were detected in the kidneys of rodents [119,120,121,122].

Among food-producing animals, pigs are the most susceptible animals to OTA exposure [10,11]. In pigs, OTA usually causes kidney disease by damaging proximal tubules [51]. It was initially described as spontaneous nephropathy in Bulgaria and was observed frequently during the carcass inspection of pigs with nephropathy problems. These pathomorphological changes in kidneys resemble those in mycotoxic porcine nephropathy observed in South Africa, both of which have a multi-mycotoxic etiology and are differentiated from mycotoxic porcine nephropathy described in Denmark [123].

In humans, even though the epidemiological evidence was inadequate, OTA exposure was associated with Balkan Endemic Nephropathy (BEN), Chronic Interstitial Nephropathy (CIN) and other kidney diseases [124,125]. ΒΕΝ, firstly recognized in the 1950s [126], was characterized by progressive kidney and urinary tract tumors in people living in the Balkans, mainly in Bulgaria, Romania and former Yugoslavia [127,128]. However, later studies have associated epigenetic changes as a possible cause of the disease [129,130]. The identification of the causative factors of BEN seems to be difficult. New scientific findings are constantly emerging, and toxic compounds produced by other fungi such as *P. polonicum*, *P. aurantiogriseum* and *P. commune* have been also associated with the etiology of BEN [131]. It has also been suggested that the disease is rather associated with exposure to other mycotoxins than OTA, and the chronic dietary intake of aristolochic acid has been stated to be mainly responsible for BEN [132,133].

In humans, two other forms of CIN nephropathy observed in Tunisia were associated with OTA [134]. However, recent studies revealed that CIN nephropathies found in Tunisia have no connection with OTA [14].

Apart from the kidneys, the liver is one of the major target organs of OTA biotransformation [135]. Significant lesions have also been observed in rat livers [135,136,137]. In mice, approximately 33% of OTA is eliminated through the hepatobiliary route of excretion, and the enterohepatic recirculation of the toxin in mice and rats is mainly responsible for the liver damages in these species [138]. Furthermore, OTA affects reproduction systems and fertility in animals [113].

## 5. Regulations on OTA in Pork Meat and Meat Products

There are remarkable differences in legal regulations of the presence of OTA in feeds or foods among various countries. In the USA, no limits on OTA in foods or feed have been set [34]. The US Food and Drug Administration (FDA), acting under the Federal Food, Drug and Cosmetic Act (FFDCA), requires the implementation of food safety plans in food industries and for good agricultural and manufacturing practices to be applied. In addition, certain countries such as Australia and Canada have adopted a similar approach on OTA [35].

Legislation differences exist in various countries in Asia. In China, Indonesia, Korea, Malaysia and recently in Singapore, legislative limits have been set for OTA in foods and feed, but not in meat and meat products [139]. In Japan, OTA is not regulated by defined maximum levels in foods and, in accordance with the USA, food safety guidelines have also been applied [35].

The EU, based on the scientific opinion of the scientific panel on contaminants in the food chain of the EFSA, established maximum limits for OTA in a variety of foods in Commission Regulation (EC) No. 1881/2006 [8], which is active today, although it has been repeatedly changed. Furthermore, a maximum level for OTA has been defined for some spices used in the production of meat products [8,140]; for example, 15 ng/g for white and black pepper; 20 ng/g for dired chillies, chilli powder and paprika; and 15 ng/g for mixtures containing one or more of the aforementioned spices.

However, in pork meat and meat products and other foods of animal origin, OTA limits have been not set yet. Instead, in many EU member countries, to protect consumers from OTA-contaminated pork meat, edible offal and derived products have been subject to regulations or guidelines to limit exposure, such as in Denmark (10 μg/kg in pig kidney, 25 μg/mL in pig blood), Estonia (10 μg/kg in pig liver), Romania (5 μg/kg in pig kidney, liver and meat) and Slovakia (5 μg/kg in meat), Italy (1 μg/kg in pork meat and derived products) [33]. The differences in OTA limits necessitate a harmonized approach to legally regulating OTA in pork meat, edible offal and meat products in EU countries.

## 6. Methods for the Detection and Determination of OTA

Several analytical methods have been used for the detection and measurement of OTA in feeds and foods [35,112,141,142]. However, the most common analytical methods are high-performance liquid chromatography (HPLC), enzyme-linked immunosorbent assay (ELISA) and thin-layer chromatography (TLC) [35].

Meat is a complex compound matrix and poses difficulties in OTA analysis due to the strong bonds between proteins and OTA and the presence of fat, which may be also co-extracted [143]. Therefore, for OTA analysis in protein-rich foods such as meat, acidic solvents are used to break the protein bonds [144]. Additionally, the presence of the OTA at trace levels in meat demands sensitive and accurate analytical methods for OTA detection [23].

HPLC with fluorescence detector (HPLC-FLD) has shown good analytical performance for OTA determination in pork tissues and organs, as the limits of detection (LOD) and limits of quantification (LOQ) values were found to be quite low. In pork meat, low levels of LOD at 0.01 μg/kg [145] and LOQ at 0.03 μg/kg [146] were reported by using HPLC-FLD analysis. Similarly, a low LOD at 0.0125 and LOQ at 0.0250 µg/kg were found in pork meat by using HPLC-FLD analysis [15]. Similar HPLC-FLD sensitivity was found in pork kidney and liver with the LOD and LOQ levels at 0.001 μg/kg and 0.002 μg/kg, respectively [147]. HPLC-FLD based on an immunoaffinity clean-up step, with a range of applicability of 0.4 to 12 μg/kg of OTA, was used for the quantification of OTA in kidney, liver, lung and pork-derived products [148]. The HPLC-FLD method has also been used for the determination of OTA in pig tissues [22,149] and in meat products [32,150].

The ELISA method is also an effective quantitative method for OTA screening in pork meat products [29,31,57,151]. An ELISA analysis for OTA presence was used in dry-cured meat products, cooked sausages and pork raw materials such as blood, brain, liver, kidney, adipose tissue, lungs and spleen [21,31,152].

The use of enzymatic digestion (ED) extraction significantly reduces matrix interference with the OTA in meat, leading to more reliable results of the OTA analytical methods [147,153]. For example, ED with the pancreatin method coupled to HPLC-FLD has been successfully used for the rapid analysis of OTA in pork meat and pork products [147,153,154]. Furthermore, ED use with HPLC for OTA quantification in pig muscle resulted in LOD and LOQ levels of 0.21 μg/kg and 0.70 μg/kg, respectively [155]. Several analytical procedures based on LC-MS/MS have been developed for the detection of OTA in pork meat products [13,156,157]. Furthermore, the LC-MS/MS analysis of OTA in traditional dry-fermented homemade sausages gave low LOD and LOQ values of 0.44 μg/kg and 1.44 μg/kg, respectively, indicating a high prevalence of OTA in these meat products [56]. It has been also proved that a sensitive liquid chromatography/electrospray ionization tandem mass spectrometry (LC/ESI-MS/MS) method could be used for the quantitative monitoring of OTA in pig kidney samples [158] and other pork tissues such as liver and muscle [156,157]. A new developed immunoaffinity column clean-up step (IAC)-LC–ESI-MS/MS was also used for OTA examination in pork meat samples [144].

Ultra-performance liquid chromatography (UPLC) has been also used for OTA detection in meat [159,160], with a higher OTA analytical sensitivity as compared to HPLC [161]. Brera et al. [158], comparing HPLC-FLD and ultra-performance liquid chromatography tandem mass spectrometry (UPLC-MS/MS), concluded that both methods are suitable for the detection of OTA in ham.

In order to detect potentially ochratoxinogenc fungi and quantify the genes involved in the biosynthesis of OTA production, tge quantitative real time PCR (qPCR) method was also used [27].

## 7. Occurrence of OTA in Slaughtered Pigs

In pigs, the highest concentrations of OTA are usually found in the blood, followed by the kidneys, liver, muscles and fat [16,17,18,19]. This OTA distribution trend in pig organs was also verified in pigs fed with OTA contaminated feed due to the “carry-over” effect (Table 2). The OTA levels in pig tissues were dependent on the OTA levels administrated in feed as well as the duration of feeding with OTA-contaminated feed [122,162]. Pigs fed with naturally contaminated feed showed higher OTA serum/plasma levels compared to studies in which pigs were fed OTA-contaminated feed [163].

Accomplished data on the occurrence of OTA in edible pig tissues from published studies conducted in various countries worldwide are summarized in Table 3. According to several studies of slaughtered pigs accomplished in various countries, the occurrence of OTA-positive samples was high in pig tissues, and particularly in blood serum and kidneys. For example, 98% and 94% of porcine serum samples tested by HPLC-FD in Romania in the studies of Curtui et al. [164] and Curtui and Gareis [165], respectively, were found to be OTA-positive. Curtui and Gareis [165] reported that the levels of OTA were in the range of 0.1–13.4 μg/L in porcine serum samples. In Canada, pig serum analysis revealed that all the positive samples had concentrations of OTA between 5.4 to 20 μg/L in the years 1988–1990 [166]. A lower incidence of OTA (31.1%) was found in pig serum in Serbia with OTA levels ranging between 0.22 and 220.8 μg/L [167].

The presence of OTA in kidneys is considered to be a good indicator of overall exposure of pigs to the toxin [149]. Several surveys conducted in various countries revealed that OTA occurrence in kidney samples of healthy pigs using HPLC-FD analysis was in the range from 8% [168] to 14.74% [149]. However, according to a study by Hou et al. [159], in kidney samples of healthy pigs using ultra-HPLC/MS/MS analysis, the occurrence was 87.5%. Various OTA levels in pig kidneys have been reported. In Italy, OTA levels in kidneys were in the range of 0.17–0.91 μg/kg [147] and 0.07–3.23 μg/kg [153] for pigs and wild boars, respectively. In the Czech Rebublic, Skarkova et al. [168] reported OTA levels of 0.15–0.46 μg/kg in pig kidneys. In China, the OTA concentrations ranged between 0.03 to 0.323 μg/kg in pig kidneys [160]. However, Polovinski-Horvatovic et al. [148] found OTA levels in examined pig kidneys as high as 3.97 µg/kg in Serbia. In a recent study in Belgium, 37.3% of kidney samples were OTA-contaminated at the mean level of 0.22 ± 0.25 μg/kg (up to 1.91 μg/kg) [169].

Various levels of OTA have been found to be present in the livers or muscles of slaughtered pigs. In the livers of pigs, OTA levels as high as 100% and 33% were found by using HPLC-FD or LC-MS/MS, respectively [147,153]. OTA levels in pig liver of 1.46 μg/kg [156] and 0.10–3.65 μg/kg [23] were reported in China and France, respectively. In Italy, Giacomo et al. [146] reported OTA concentrations in pork liver samples with a range of 0.07–0.59 μg/kg and a mean value of 0.35 μg/kg. Similarly, in Italy, Luci et al. [152] found that OTA ranged from 0.02 to 1.93 μg/kg in livers of wild boars. The levels of OTA in muscles of slaughtered pigs were 8% in the Czech Rebublic [168] and 33.33% [56] in China. High percentages of OTA presence in muscles of slaughtered pigs in France were found with a range of 76% to 100% [23]. Luci et al. [152] examined 48 muscle samples of slaughtered pigs in Italy, and all of them were positive for OTA presence. Pig muscles were found to be contaminated with OTA with a range of 0.15 to 0.20 μg/kg in the Czech Republic [168], 0.09–0.20 μg/kg in Italy [146] and 0.03–0.23 μg/kg in Canada [170]. In a recent study conducted in Italy, Meucci et al. [15] found that the maximum OTA concentrations in pig muscles reared in an indoor system were 0.055 μg/kg and 0.078 μg/kg in indoor and outdoor systems, respectively. Analysis by LC-MS/MS showed higher OTA levels in muscles of 1.25 μg/kg in China [156] and 0.88 μg/kg in Italy [171]. Analysis by SIDA–UHPLC–MS/MS showed also high levels of OTA (maximum 1.15 μg/kg) in the muscles of French pigs from organic farming production systems [23]. The OTA levels in the fat of slaughtered pigs in Italy were found to be low, with values ranging 0.079 ± 0.018 μg/kg and 0.085 ± 0.025 μg/kg for indoor and outdoor systems, respectively.

**Table 2 toxins-14-00067-t002:** Distribution of OTA in tissues of pigs fed with OTA-contaminated feed.

OTA Treatment	Sample		OTA Content (μg/kg-μg/L)	Method ^2^	Reference
	Type	Number			
1 male, 1 female control, 1 male, 1 female OTA-treated (0.8 μg/kg feed)/6 months	Kidneys	2 control 2 treated	12.1 and 9.6 (control) 98.3 and 103.8 (treated)	HPLC-FD	[172]
Control/OTA-treated (25 µg/kg feed)/119 days	Kidneys	32 control 32 treated	10.50 (control) 69 (treated)	HPLC-FD	[173]
	Liver		3.50 (control) 52.00 (treated)		
	Meat (*Semimembranosus* muscle)		0.88 (control) 6.10 (treated)		
OTA treated: 2.5 mg/kg feed	Kidneys	5 treated	Mean: 29.15	TLC and spectrophotometry	[174]
	Liver		Mean: 20.1		
	Meat (heart/muscle)		Mean: 12.6		
T0: control group, OTA treated groups (T1-T3): 50, 100, 200 μg/kg feed/2 weeks	Blood	24 (total)	T0: <0.02–0.26, Mean: 0.19 T1: 5.24–7.51, Mean: 6.35 T2: 7.41–16.5, Mean:11.4 T3: 17.3–34.5, Mean: 24.6	HPLC-FD LOD: 0.02, LOQ: 0.05	[175]
	Kidneys		T0: <0.04–0.32, Mean: 0.13 T1: 2.75–4.37, Mean: 3.74 T2: 4.56–5.72, Mean: 5.24 T3: 7.33–11.8, Mean:10	LOD: 0.04, LOQ: 0.10	
	Liver		T0: <0.04–0.14, Mean: 0.06 T1: 1.26–1.82, Mean: 1.60 T2: 1.91–2.56, Mean: 2.35 T3: 3.16–6.98, Mean: 4.29	LOD: 0.04, LOQ: 0.10	
	Meat		T0: ND ^1^ T1: 0.60–0.89, Mean: 0.74 T2: 1.08–1.45, Mean: 1.27 T3: 1.67–3.40, Mean: 2.23	LOD: 0.04, LOQ: 0.10	
	Fat		T0: ND T1: 0.57–0.79, Mean: 0.68 T2: 0.86–1.26, Mean: 1.04 T3: 1.33–2.58, Mean: 1.71	LOD: 0.02, LOQ: 0.05	
5 controls, 5 OTA treated (0.78 mg/day) (300 μg/kg feed)	Blood	10 (total)	ELISA Mean: 6.56 ± 2.15 HPLC-FD Mean: 6.35 ± 2.49	ELISA: LOD: 1.34, LOQ: 2.94 HPLC-FD LOD: 0.15, LOQ: 0.20	[21]
	Kidneys		ELISA Mean: 14.59 ± 3.47 HPLC-FD Mean: 15.31 ± 3.11	ELISA: LOD: 1.59, LOQ: 3.32 HPLC-FD: LOD: 0.15, LOQ: 0.20	
	Liver		ELISA Mean: 8.23 ± 2.49 HPLC-FD Mean: 8.81 ± 2.08	ELISA: LOD: 2.31, LOQ: 5.67 HPLC-FD: LOD: 0.15, LOQ: 0.20	
	Muscle		ELISA Mean: 5.42 ± 1.13 HPLC-FD Mean: 5.61 ± 2.01	ELISA: LOD: 0.39, LOQ: 0.57 HPLC-FD: LOD: 0.15, LOQ: 0.20	
	Fat		ELISA Mean: 4.31 ± 1.58 HPLC-FD Mean: 4.59 ± 1.68	ELISA: LOD: 0.32, LOQ: 0.40 HPLC-FD: LOD: 0.15, LOQ: 0.20	
OTA treated (250 μg/kg feed)/4 weeks	Blood	5 (total)	3.71–6.57 Mean: 4.77 ± 1.57	ELISA: LOD: 0.20, LOQ: 0.31 HPLC-FD: LOD: 0.10, LOQ: 0.15	[176]
	Kidneys		11.88–15.98 Mean: 13.87 ± 1.41	ELISA: LOD: 1.44, LOQ: 1.89 HPLC-FD: LOD: 0.24, LOQ: 0.36	
	Liver		4.89–9.78 Average: 7.28 ± 1.75	ELISA: LOD: 1.54, LOQ: 2.11 HPLC-FD: LOD: 0.36, LOQ: 0.42	
	Meat		2.79–5.37 Mean: 4.72 ± 0.86	ELISA: LOD: 0.45, LOQ: 0.61 HPLC-FD: LOD: 0.16, LOQ: 0.22	
	Fat		2.95-5.26 Average: 4.11 ± 0.88	ELISA: LOD: 0.66, LOQ: 1.11 HPLC-FD: LOD: 0.23, LOQ: 0.29	

^1^ ND: Not detected. ^2^ HPLC-FD: High-performance liquid chromatography with fluorescence detector; TLC: Thin layer chromatography; ELISA: Enzyme-linked immunosorbent assay; LOD: Limit of detection (μg/kg-μg/L); LOQ: Limit of quantification (μg/kg-μg/L).

**Table 3 toxins-14-00067-t003:** Occurrence of OTA in meat and edible offal of slaughtered pigs.

Samples	Country	Year/Years of Study	OTA Prevalence		Method ^2^	Comments	Reference
			Positive/Number Tested (% Positive)	Concentration (μg/kg-μg/L)			
Blood	Canada	1988	1200 Total (3.6% of 194) (4.2% of 1006)	>20	HPLC-FD		[166]
		1989–1990	16–65%	5.4–19.4			
Serum	Bulgaria	1993–1994	25/75 (48–64%, autumn) (60.88 spring)	Mean: 4.8–21.94 (autumn) 60.88 (spring)	HPLC		[177]
	Romania	1998	52 (Total)	98%: 0.05–13.4 92%: ≥0.1 Max: 13.4 Mean: 2.43	HPLC-FD	LOD: 0.1	[164]
	Romania	NR ^1^	49/52 (94%)	0.1–13.4	HPLC-FD	LOD: 0.1	[165]
	Serbia	2006–2007	28/90 (31%)	0.22–220.8 Mean: 3.70 ± 23.59	HPLC-FD	LOD: 0.1	[167]
Kidneys	Bulgaria	1994	80–100% (nephropathic kidneys)	Mean: 1.5–7.17		Samples from porcine nephropathy affected farms	[178]
	France	1997	3/300 (1%) 6/100 (6%)	1%: 0.40–1.40 6%: 0.16–0.48	HPLC-FD	300 Healthy pigs 100 Nephropathic pigs LOD: 0.05, LOQ: 0.16	[179]
	France	1998	238/710 (33.5%)	184/710 (25.9%): LOD-0.5 54/710 (7.6%): 0.5–5	HPLC-FD	LOD: 0.05, LOQ: 0.16	
	Germany	NR	26/58 (44%)	Max: 9.3	HPLC-FD	LOD: 0.01	[180]
	Romania	1998	41/52 (79%)	Max: 3.18, Mean: 0.54	HPLC-FD	LOD: 0.01	[165]
	Denmark	1999	284/300 (94.7%)	0–15 Mean: 0.50, Median: 0.18	HPLC-FD	LOD: 0.02, LOQ: 0.06	[181]
	Italy	NR	52/54 (96%)	0.26–3.05	HPLC-FD	LOD: 0.14, LOQ: 0.52	[182]
	Italy	NR	54/54 (100%)	Mean: 0.29, Max: 0.9	ELISA	LOD: 0.01	[183]
	Italy	2005	5 (Total)	23.9–27.5 Average: 25.6 ± 1.56	HPLC-FD	LOD: 0.10, LOQ: 0.30	[184]
	Serbia	2006–2007	30/90 (33.3%)	0.17–52.5 Mean: 1.26 ± 5.85	HPLC-FD	LOD: 0.01	[167]
	Czech Rebublic	2011–2012	8%	0.15–0.46, Mean: 0.18	HPLC-FD	LOD: 0.10, LOQ: 0.30	[168]
	China	2014	35/40	(0.03–0.1) to 0.323	UHPLC-MS/MS	LOD: 0.03, LOQ: 0.10	[159]
	Italy	NR	5/5 (100%)	0.17–0.91 Mean: 0.37 ± 0.30	HPLC-FD (ED)	LOD: 0.001, LOQ: 0.002	[147]
	Belgium	2012–2019	41/110 (37.3%)	Mean: 0.22 ± 0.25	LC-MS/MS	LOD: 0.2	[169]
Kidneys of wild boars	Italy	2014–2015	48 (Total) 2014:26/26 (100%) 2015:22/22 (100%)	2014: 0.19–3.23 Median: 0.68 2015: 0.07–1.72 Median: 0.34	HPLC-FD (ED)	LOD: 0.001, LOQ: 0.002	[120]
	Serbia	2018	14/95 (14.74%)	0.10–3.97 Average: 1.36 Median: 0.99	HPLC-FD	LOQ: 0.10	[148]
Liver	Germany	NR	10/58 (17%)	Max: 2.7 µg/kg	HPLC-FD	LOD: 0.01	[180]
	Romania	1998	39/52 (75%)	Max: 0.61 Mean: 0.16	HPLC-FD	LOD: 0.01	[165]
	Italy	2005	5 (Total)	3.2–5.3 Average: 4.4 ± 0.8	HPLC-FD	LOD: 0.10, LOQ: 0.30	[184]
	Serbia	2006–2007	24/90 (26.6%)	0.22–14.5 Mean: 0.63 ± 1.87	HPLC-FD	LOD: 0.01	[167]
	China	NR	1/3 (33.33%)	1.46	LC-MS/MS	LOQ: 0.25-1.0	[156]
	Italy	NR	5/5 (100%)	0.07–0.59 Mean: 0.35 ± 0.20	HPLC-FD (ED)	LOD: 0.001, LOQ: 0.002	[147]
Liver of wild boars	Italy	2014–2015	48 (Total) 2014: 26/26 (100%) 2015: 22/22 (100%)	2014: 0.04–1.93, Median: 0.15 2015: 0.02–1.31 Median: 0.23	HPLC-FD (ED)	LOD: 0.001, LOQ: 0.002	[153]
	France	2014	47/70 (67%)	0.10–3.65	SIDA–UHPLC–MS/MS	LOD: 0.03, LOQ: 0.10	[23]
Pork meat and liver	Denmark	1993–1994	64/76 (84.2%) (conventional) 4/7 (57.1%) (ecological)	Conventional Max: 1.3 Mean: 0.11, Median: 0.09 Ecological Max: 0.12, Mean: 0.05, Median: 0.05	HPLC-FD	LOD: 0.02–0.03	[185]
Meat	Germany	NR	10/58 (17.2%)	Max: 0.14, Median: <0.01	HPLC-FD	LOD: 0.01	[180]
	Romania	1998	9/54 (17%)	Max: 0.53, Mean: 0.15	HPLC-FD	LOD: 0.01	[165]
	Denmark	1999	228/300 (76%)	0–2.9 Mean: 0.12, Median: 0.03	HPLC-FD	LOD: 0.03, LOQ: 0.09	[181]
Swine muscle	Portugal	2002–2003	1/13 (7.7%)	0.12 Mean: 0.01 ± 0.03	HPLC-FD	LOD: 0.01, LOQ: 0.04	[144]
	Italy	NR	54/54 (100%) 42/54 (78%) > 0.05 20% > 0.5	Mean: 0.024, Median: 0.01	ELISA	LOD: 0.01	[183]
	China	NR	1/3	1.25	LC-MS/MS	LOQ: 0.25–1.0	[156]
	Czech Rebublic	2011–2012	8%	0.15–0.20, Mean: 0.13	HPLC-FD	LOD: 0.10, LOQ: 0.30	[168]
	Italy	NR	5/5 (100%)	0.09–0.20, Mean: 0.13 ± 0.04	HPLC-FD (ED)	LOD: 0.001, LOQ: 0.002	[146]
Muscle of wild boars	Italy	2014–2015	48 (Total) 2014: 26/26 (100%) 2015: 22/22 (100%)	2014: <LOD–0.77 Median: 0.08 2015: 0.03–0.50 Median: 0.13	HPLC-FD (ED)	LOD: 0.001, LOQ: 0.002	[152]
	China	NR	1/4	0.88	LC-MS/MS	LOD: 0.07, LOQ: 0.25	[156]
	France	2014	19/25 (76%)	≤0.03–1.15	SIDA–UHPLC–MS/MS	LOD: 0.03, LOQ: 0.10	[23]
	Italy	NR	5/5 (100%) Rearing system: Indoor & Outdoor	Indoor: 0.055 ± 0.015 Outdoor: 0.078 ± 0.011	HPLC-FD	LOD: 0.0125, LOQ: 0.0250	[15]
Fat	Italy	NR	5/5 (100%) Rearing system: Indoor & Outdoor	Indoor: 0.079 ± 0.018 Outdoor: 0.085 ± 0.025	HPLC-FD	LOD: 0.0125, LOQ: 0.0250	[15]

^1^ NR: Not reported. ^2^ Refer to Table 2 for abbreviations; UHPLC-MS/MS: Ultra-high performance liquid chromatography tandem mass spectrometry; LC-MS/MS: Liquid Chromatography tandem mass spectrometry; ED: enzymatic digestion; SIDA: Stable isotope dilution assay.

## 8. Occurrence of OTA in Pork Meat Products

OTA is the most common mycotoxin found in processed pork meat products [30,67,70,83]. Table 4 summarizes OTA occurrence data in pork meat products from published studies conducted in various countries. The contamination of pork meat products with OTA may be due to the use of contaminated raw meat and offal, especially kidneys, livers, blood and blood plasma [21,25,26], and secondarily to the addition of other contaminated materials such as spices [31,32]. Offal-containing sausages such as black pudding and liver sausages have been often found to be contaminated with OTA in significant concentrations [21]. However, in a recent study in Belgium, 20 black sausage samples were tested and were not found to be contaminated with OTA [169].

High differences in the occurrence of OTA in pork meat products have been observed in various countries (Table 4). High values of OTA of 158 μg/kg and 103.69 μg/kg were reported for fermented sausages in Denmark and salami in Italy, respectively [142,186]. Mitchell et al. [187] reported that most pork meat samples were found to have non-detectable OTA levels. According to a survey conducted in various European countries, the mean value of OTA recorded in various pork products was 0.052 μg/kg [188]. In Italy, Altafini et al. [32] analyzed 172 different salamis and found that 3 samples of spicy salamis exceeded the official Italian permitted value of 1 μg/kg allowed for OTA in pork meat products. The authors concluded that the high OTA levels in these samples were derived from the addition of OTA-contaminated chili pepper. Meucci et al. [15] compared the effect of the indoor and outdoor rearing system in OTA contamination in produced pork meat products. The OTA values determined by HPLC-FD analysis were 0.058 μg/kg and 0.537 μg/kg in indoor and 0.064 μg/kg and 0.558 μg/kg in outdoor salami and mortadella, respectively, indicating no significant difference between the two examined systems.

Studies on the OTA levels in fermented sausages produced from pigs fed with OTA-contaminated feed were also conducted. In a study conducted in Croatia, pigs were fed with OTA-contaminated feed (300 μg/kg of feed) for 30 days [21]. The mean OTA levels were 13.82 μg/kg and 9.13 μg/kg for produced black pudding sausages and pated products, respectively. In a similar study conducted also in Croatia, the OTA levels were ranged from 4.51 ± 0.11 μg/kg in smoked ham to 6.32 ± 0.65 μg/kg in bacon [152].

The occurrence of OTA in contaminated meat products may also be due to the growth of OTA-producing fungi in these products [26,27,28,29,30,94]. Surveys in ham-manufacturing plants revealed that OTA toxigenic strains of Aspergilli and Penicillia were present in the ripening rooms [98,189]. Dall’Asta et al. [26] examined the contamination levels of hams in the inner and the outer parts and concluded that the mean OTA levels were 0.24 μg/kg in the inner and 0.98 μg/kg in the outer samples. Hams inoculated with *P. nordicum* yielded significantly higher amounts of OTA than those inoculated with *A. ochraceus* [27]. Sánchez-Montero et al. [24] inoculated ham samples with *P. verrucosum* and *P. nordicum* and found that the OTA levels ranged between 2.30–4.37 μg/kg, using different a_w_ values. In a recent study, Delgado et al. [161] inoculated raw sausages with *P. nordicum,* and after 26 days of ripening, the OTA levels were in the range of 1.02 to 51.06 μg/kg.

It was also found that the growth of *A. westerdijkiae* on the salami surface produces high levels of OTA on the casing and allows its diffusion through the casing to the outer parts of sausages [30]. Furthermore, the inoculation of dry fermented sausage with *A. westerdijkiae* resulted in high levels of OTA of 1.959 μg/kg [70]. In sausages inoculated with strains of *P. nordicum* and *P. verrucosum*, the OTA level was found to be between 30.58 μg/kg and 66.91 μg/kg [24]. Fresh pork sausages were inoculated with *P. nordicum*, and OTA was detected on the fourth day (10 μg/kg) and increased significantly on the seventh day, reaching the maximum level of 135 μg/kg after 10 days of storage [67].

**Table 4 toxins-14-00067-t004:** OTA occurrence in pork meat products.

Samples	Country	Year/Years of Study	OTA Prevalence		Method ^1^	Comments ^1^	Reference
			Positive/Number Tested (% Positive)	Concentration (μg/kg)			
Various Products	Various countries	1990–1998	NR	Mean: 0.052	NR		[188]
	France, Germany, Italy, U.S.	1997–1999 and 2000–2002	18%	NR	NR		[190,191]
Liver sausages	Germany	NR ^1^	68%	Mean: 0.02, Max.: 4.56	HPLC-FD		[180]
Bologna type products			46.7%	Mean: 0.01, Max.: 0.38			
Blood sausages			77.2%	Mean: 0.04, Max.: 3.16			
Salami	Italy	2001–2002	4/12 (33%)	Mean: 0.02, Max.: 0.08	HPLC-FD	LOD: 0.01, LOQ: 0.03	[145]
Cooked ham			1/12 (8%)	Mean: 0.004, Max.: 0.05			
Dry-cured ham			12/30 (40%)	Mean: 1.62, Max.: 28.42			
Coppa			5/18 (28%)	Mean: 0.03, Max.: 0.24			
Würstel			1/12 (8%)	Mean: 0.005, Max.: 0.06			
Hams	Italy	NR	Inner samples: 2/10 (20%) Outer samples: 5/10 (50%)	0.28–1.52, 0.11–7.28	HPLC-FD	LOD: 0.02, LOQ: 0.06	[192]
		NR	Inner samples: 32/110 (29%) Outer samples: 84/110 (76.4)	4.66–12.51	HPLC-FD	LOD: 0.1, LOQ: 0.3	[26]
Dry-cured hams	Italy	2007–2010	NR	Means: 0.6–4.11	HPLC-FD	Choroform Extraction: LOD: 0.090, LOQ: 0.180	[190]
				Means: 1.14–6.29		Enzyme Assisted extraction: LOD: 0.060, LOQ: 0.120	
Fermented sausages and dry-cured hams	Denmark	NR	1/22 (4.5%)	Positive sample: Parma ham 1st analysis: 56, 2nd: 158 and 113	LC-MS/MS	LOD: 11, LOQ: 50	[142]
Ham	Croatia	2011–2014	18/105 (17.14%)	0.97–9.95, Means: 0.16–1.82	ELISA HPLC-FD	ELISA: LOD: 0.85–0.98 LOQ: 1.56–1.95 HPLC: LOD: 0.15, LOQ: 0.20	[31]
Dry-fermented sausages			14/208 (6.73%)	0.95–5.10, Means: 0.08–0.21			
Bacon			2/62 (3.22%)	ND-1.23, Means: ND-0.07			
Cooked sausages			3/35 (8.57%)	ND-3.13, Means: ND-0.26			
Dry-fermented sausages (industrial)	Croatia	2013	18/56 (32.1%)	1.36–7.12, Mean: 3.02 ± 2.45	ELISA	Sausages LOD: 0.84, LOQ: 1.07 Ham LOD: 0.32, LOQ: 0.40	[150]
Dry-fermented sausage (homemade)			11/77 (14.3%)	1.36–6.26, Mean: 3.54 ± 1.70			
Dry-cured ham			12/54 (22.1%)	1.56–9.95, Mean: 3.16 ± 2.42			
Salami	Italy	NR	NR Rearing system indoor and outdoor	Indoor: 0.058 ± 0.015 Outdoor: 0.064 ± 0.004	HPLC-FD	LOD: 0.0125, LOQ: 0.0250	[15]
Mortadella				Indoor: 0.537 ± 0.042 Outdoor: 0.558 ± 0.016			
Prosciuttos	Croatia	NR	15/67 (22.4%)	2.16–6.86, Means: 3.56–5.04	ELISA		[57]
Fermented sausages			7/93 (7.5%)	2.74–4.14, Means: 2.97–3.89			
Hams	Italy	NR	27/42 (64.2%) <1.0 14/42 (33.4%) 1–2 1/42 (2.4%) > 2	0.04–0.98, 1.1–2, 2.20–2.30	HPLC-FD VICAM fluorometer	HPLC: LOD: 0.04 Fluorometric: LOD: 0.7	[171]
Salami type cured meat	Italy	NR	14/30 (46.7%)	9/30 (30%): 0.006–0.06 5/30 (16.7%): 0.06–1	HPLC-FD	LOD: 0.06, LOQ: 0.22	[193]
Salami	Italy	2013	5/50 (10%)	4 samples: 0.06–0.44 1 sample: 103.69	HPLC-FD	LOD: 0.06, LOQ: 0.20	[186]
		2013–2015	13/133 (9.8%)	-	LC-MS/MS	LOQ: 1	[194]
		2015–2016	22/172 (12.8%)	0.07–5.66, Mean: 0.51	HPLC-FD	LOD: 0.05, LOQ: 0.20	[32]
Sausages	China	2013–2014	1/10	0.5	LC-MS/MS	LOD: 0.05, LOQ: 0.1	[13]
Dry Fermented sausages	Croatia	NR	13/88 (14.8%)	<LOD-0.48, Mean: 0.26 ± 0.12	LC-MS/MS	LOD: 0.44, LOQ: 1.44	[56]
Cured sausages	Italy	NR	72/160 (45%)	Cases of sausages: 3–18 Means: 4.5–8.0	ELISA	ELISA: LOD: 0.1	[94]
“Pâté” products	Spain	NR	3/38 (7.9%)	Max.: 1.77	HPLC-FD	LOD: 0.56, LOQ: 0.84	[195]
Products with porcine serum	Germany	NR	58/325 (17.8%)	Mean: 0.15	ELISA and HPLC-FD	-	[196]

^1^ Refer to Table 3 for abbreviations.

## 9. Risk Assessment of Human Exposure to OTA by Consumption of Pork Meat and Derived Products

There are only a few studies on the assessment of the human exposure to OTA due to the consumption of pork meat and derived products. The Joint FAO/WHO expert committee on food additives (JECFA) established a Provisional Tolerable Weekly Intake (PTWI) of 100 ng/kg bw based on the lowest-observed-adversed-effect level (LOAEL) for renal effects in pigs [188,197]. EFSA has adopted a scientific opinion relating to OTA and derived a tolerable weekly intake (TWI) of 120 ng/kg bw (equivalent to 17 ng/kg bw/day), derived from the pig lowest observed adverse effect level (LOAEL) [8], and this was also reconfirmined in 2010 [198]. Health Canada re-evaluated the appropriateness of the EFSA TWI and established a tolerable daily intake (TDI) of 3 ng/kg bw per day (which would correspond to a TWI of 21 ng/kg bw) [14]. In the European population, dietary exposure levels for adult consumers have estimated a range from 15 to 60 ng/kg bw per week (ca. 2 to 8 ng/kg bw per day), and this range is below the established TWI. However, the exposure could be higher in children and certain cosumer groups with specific consumption habits [97,199].

According to the results of two research projects (1997–1999 and 2000–2002) [190,191] on the occurrence of OTA in commodities on the European market and on dietary exposure to OTA in the EU members, pork contributed only 1% to the total estimated intake. In an early study of Frohlich et al. [166], OTA was present in the blood of people in Canada (40%), and they concluded that a possible entry path of OTA into the human food chain was pork products. Frank [196] calculated that the daily intake of OTA by German consumers is 1.6 ng from the consumption of sausages. In a recent study in USA, mean OTA exposure from pork in the consumer population was 0.16 ng/kg bw per day [187]. JECFA [188] estimated the intake of OTA from pork to be 1.5 ng/kg bw per week. Jorgensen [200] estimated the OTA exposure by European consumers at 45 μg/kg bw per week and the intake of OTA from pork at 1.5 ng/kg bw per week.

Human exposure to OTA seems to be associated predominantly with the consumption of contaminated plant-derived products and only to a minor extent with foods of animal origin [201]. However, regular consumption of certain porcine blood products contributes considerably to the level of exposure, especially in children, in which the relatively lower body weight as compared to adults results in a higher exposure per kg bw [202].

## 10. Conclusions

OTA is produced by grown *Aspergillus* and *Penicillium* spp. in a wide variety of foods and feeds. OTA is toxigenic to animals and humans and has been classified as possibly carcinogenic to humans (Group 2B) by the IARC. In humans, OTA exposure was associated with Balkan Endemic Nephropathy, Chronic Interstitial Nephropathy and other kidney diseases, although this is not epidemiologically evident.

Pigs are the most susceptible animals to OTA exposure. Among foods of animal origin, pork meat and meat products are considered as important sources for chronic dietary exposure to OTA in humans. The EU established maximum limits for OTA in a variety of foods since 2006, but not for meat or meat products.

Various levels of OTA have been found to be present in pork meat and offal from slaughtered pigs in various countries. According to several studies of slaughtered pigs accomplished in many countries, the OTA levels were particularly high in blood serum and kidneys.

High differences in the occurrence of OTA in pork meat products have been observed in various countries. Pork products made from pig blood or organs such as the kidney or liver have often been associated with the presence of OTA. Results from these studies highlight the need to apply control OTA measures in pig feeds and establish an ML for OTA in pork meat by-products to protect human health and to constantly monitor OTA occurrence in animal-derived products.

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
