# Peer review of "Ochratoxin A in Slaughtered Pigs and Pork Products"

_toxins, 2022, doi:10.3390/toxins14020067_

Round 1
Reviewer 1 Report
Dear authors,
I thorouhgly read your manuscript and have a few recomendations.
Line 74-78
Latin names are not in italics.
Section 3 and 4
Should be moved before section 2 or partially included to introduction section. In current layout the text and main idea is fragmented and it distracts the reader from the topic. For this review it would be better firstly generally summarize information about OTA and than flood the reader by information of main topic.
Table 4
reference 11 - Salami - should be in midle of cell
I very much appreciate the approach of the authors, especially in the area of information submitted in the form of now well-arranged tables
Author Response
Response to the reviewers’ comments on the manuscript entitled
“Ochratoxin A in slaughtered pigs and pork products”
(Manuscript ID Toxins-1498025)
Response to Review Report 1
The authors would like to thank the reviewer for his valuable comments that helped us to improve the quality of our manuscript.
The manuscript was revised according to reviewer’s 1 comments. The revised text was marked in yellow color.
Reviewer 1
Dear authors,
I thorouhgly read your manuscript and have a few recomendations.
Line 74-78
Latin names are not in italics.
Response: Your comment was adopted (lines 74-77 of revised version of the manuscript).
Section 3 and 4
Should be moved before section 2 or partially included to introduction section. In current layout the text and main idea is fragmented and it distracts the reader from the topic. For this review it would be better firstly generally summarize information about OTA and than flood the reader by information of main topic.
Response: The presentation of these sections in the present form was asked by previous reviewers. The text was revised and some of the information of the original version of manuscript was removed (lines 151-159 of original version were deleted).
Table 4
reference 11 - Salami - should be in midle of cell
Response: Your comment was adopted, and the text was revised accordingly.
I very much appreciate the approach of the authors, especially in the area of information submitted in the form of now well-arranged tables
Response: Thank you for your comment.

Reviewer 2 Report
Dear Author,
Overall manuscript is ok.
Add few figures summarizing the various effect of Ochratoxin on human health. Also, effects on different organ.
Add detections techniques for the OTA.
Author Response
Response to the reviewers’ comments on the manuscript entitled
“Ochratoxin A in slaughtered pigs and pork products”
(Manuscript ID Toxins-1498025)
Response to Review Report 2
The authors would like to thank the reviewer for his valuable comments that helped us to improve the quality of our manuscript.
The manuscript was revised according to reviewer’s 2 comments.
Dear Author,
Overall manuscript is ok.
Add few figures summarizing the various effect of Ochratoxin on human health. Also, effects on different organ.
Response: The text is quite long to include such figures, which may be interesting.
Add detections techniques for the OTA.
Response: Please see section 6 of revised version of the manuscript.

Reviewer 3 Report
General comments and MAJOR CONCERNS:
This review paper suffers from unsufficient study in the available literature in regard to OTA contamination in pork meat, kidney or blood in various European countries and from some inaccuracies in the quoted data, and therefore a major revision should be done.
Part of information given in some chapters such as “3. Physicochemical properties of OTA” or “6. Methods for the detection and determination of OTA” is well known and not in line of the main focus of this paper – see the specific comments. Such information is not quite helpful for clarifying the way of OTA contamination of cereal and cereal by products, which are important ingredients in pig feeds and are responsible for OTA contamination in pork meat and offal – see the specific comments. The focus in such chapters should be only on the suggested methods and procedures for the detection of OTA in pork meat by-products.
The possible preventive measures to decrease human exposure from pork meat or offal from slaughtered pigs should be suggested in the Conclusion or in the Chapter 9 Risk assessment in this paper in order to increase its usefulness for readers.
Also, some of the given information in this paper is not quite correct and even wrong, e.g. the statement that Bulgarian mycotoxic porcine nephropathy is different from that described in South Africa, because just the oposit is true – it is very similar to it and both nephropathies Bulgarian and South African have multi-mycotoxic etiology in contrast to the classical porcine nephropathy as described in Denmark.
In addition, there are many inaccuracies in the thoughts expression or wrong using of unappropriate terms such as “OTA nfection of proximal tubules of kidneys” or “OTA infected pigs”, etc. It is well known that OTA is not an infectious agent, but mycotoxin, and therefore, it cannot infect the proximal tubules or the pigs.
There are also some contradictions in the “Conclusions” of this paper, where the authors state that “Human exposure to OTA seems to be associated predominantly with the consumption of contaminated plant-derived products, and only to a minor extent to foods of animal origin…” and, thereafter, in the second paragraph of the “Conclusions” there is just the oposit statement that “Pork meat and meat products are considered as important sources to the chronic dietary exposure to OTA in humans.”
Specific comments:
-Page 1, line 4 and lines 26-27 – It should added, that OTA was also found to possess carcinogenic effect in animals and poultry, in addition to its nephrotoxic, hepatotoxic, teratogenic, neurotoxic, genotoxic and immunotoxic effects in animals [see Stoev, 2015, 2020, 2021] in the sentence “Several studies have indicated the nephrotoxic, immunotoxic, teratogenic, embryotoxic, genotoxic AND CARCINOGENIC properties of OTA….”
-Stoev, S. D., Studies on carcinogenic and toxic effects of ochratoxin A in chicks, Special issue “Ochratoxins”, Toxins, 2010a, 2, 649-664
-Stoev, S.D., Long term preliminary studies on toxic and carcinogenic effect of individual or simultaneous exposure to ochratoxin A and penicillic acid in mice, Toxicon, 2020, 184, 192–201
-Stoev, S.D., Follow up long term preliminary studies on carcinogenic and toxic effects of ochratoxin A in rats and the putative protection of phenylalanine, Toxicon, 2021, 190, 41-49
-Page 1, line 31-32 – in the sentence “OTA found in consumed feedstuffs, can adversely affect the animal health, increase susceptibility to microbial infections…” it should be added that the same mycotoxin could also increase susceptibility to secondary bacterial infections in growing pigs and immunosuppression is the first expressed toxic effect of ochratoxin A (Stoev et al, 2000).
-Stoev, S. D., D. Goundasheva, T. Mirtcheva, P. G. Mantle, Susceptibility to secondary bacterial infections in growing pigs as an early response in ochratoxicosis, Experimental and Toxicologic Pathology, 2000, 52, 287-296
-Page 2, line 79-82 The sentence “OTA toxigenic A. carbonarius has been usually identified in grapes or wine [34,65-79 69], coffee beans and cocoa [34,70-74], maize, peanuts, paprika [75,76] and dried chilli 80 [58]. A. niger are isolated from several foods such as raisins, grapes, maize, peanuts and 81 coffee [73,77-80]” is not in line of the focus of this paper to clarify the way of OTA contaminated of cereal and cereal by products, which are important ingredients in pig feeds and are responsible for OTA contamination in pork meat and offal.
-Page 2, line 85-87 – In addition to the given Penicillia species of “…P. chrysogenum, P. glycyrrhizacola, P. polonicum [84], P. brevicompactum, P. crustosum, P. olsonii, P. oxalicum 86 [85], P. nalgiovense, P. solitum and P. salamii [86]…..”, the strain P. commune was also reported to produce OTA [157].
-Page 4, line 145-160 – The physicochemical properties and the biotransformation of OTA in different animals described in the chapter “3. Physicochemical properties of OTA” are well known and it is not necessary to describe them again and again.
-Page 5, line 165-167 – In the sentence “Innovative food-processing methods such as irradiation, cold plasma (CP), high‐pressure processing (HPP) may be an effective means of OTA decrease in foods [139,140]” it should be clarified that gamma or electron beam irradiation is mainly used to control the growth of ochratoxinA-producing fungi, but not to destroy OTA itself.
-Page 5, line 173-175 – In the sentence “……and can cause tumors in organs like kidneys and liver, both in animals and humans [1,2,3,31,142]…” some other organs and tissues should be added such as intestine, ureters, lung, oculi and muscles [see the papers above Stoev, 2015, 2020, 2021]
-Page 5, line 175-178 - In the sentence “Consumption of OTA contaminated feed can adversely affect the animals’ health and has been associated with several animal diseases, including, porcine nephropathy, avian ochratoxicosis [143,144] and carcinogenicity in rodents and poultry [145]” the previous 3 publications should be added here – see above.
-Page 5, line 193 and 195 – The sentence “…..OTA is usually causing kidney disease by infecting proximal tubules…” and the sentence “….observed frequently during the carcass inspection of infected pigs….” are not quite correct, because OTA is not an infectious agent, but mycotoxin, and therefore, it cannot infect the proximal tubules or pigs. These sentences should be corrected to: “…..OTA is usually causing kidney disease by DAMAGING proximal tubules…..”. and “….observed frequently during the carcass inspection of pigs WITH NEPHROPATHY PROBLEMS”….”
-Page 5, line 195-198 The sentence “It is differentiated morphologically from the classical mycotoxic porcine nephropathy described in Denmark and South Africa, both of multi-mycotoxic etiology, involving several mycotoxins including OTA as well as penicillic acid [158]” is not quite correct, because Bulgarian nephropathy is not different from this one described in South Africa, but just the oposit – it is very similar to it and both nephropathies Bulgarian and South African have multi-mycotoxic etiology in contrast to the classical porcine nephropathy as described in Denmark.
-Page 6, line 216 – In this paragraph it should be explained that in mice, approximately 33% of OTA is eliminated through the “hepatobiliary route of excretion and enterohepatic recirculation of OTA” in mice and rats is mainly responsible for the liver damages in these species [131]. The quotation of reference “[137]” is not appropriate and adequate to be given here.
-Page 6, line 233 – The maximum permited content of mycotoxins in human food and animal feed are given not only in EC Regulation No 1881/2006, but also in EC Directive 2002/32/EC, and EC Recommendations 2006/576/EC and 2013/165/EU) (see the review paper Stoev, 2015 ).
-Stoev, S. D. Foodborne mycotoxicoses, risk assessment and underestimated hazard of masked mycotoxins and joint mycotoxin effects or interaction, Environmental Toxicology and Pharmacology, 2015, 9, 794–809.
-Page 6-7, lines 246-303 – The most of information given in chapter “6. Methods for the detection and determination of OTA” is well known and not in line of the main focus of this paper, and therefore, it should be omitted or the focus should be only on the suggested methods and procedures for the detection of OTA in pork meat by-products.
-Page 7, line 305-308 The studed relationships between the levels of OTA in fed forages and blood of pigs (Stoev et al, 2001) could be mentioned in Table 2, in addition to the possible relationships between the levels of OTA in blood and the continuance of OTA exposure or the age of the pigs (Stoev et al, 2002).
-Stoev, S.D., Vitanov, S., Anguelov, G., Petkova-Bocharova, T., Creppy, E. E. Experimental mycotoxic nephropathy in pigs provoked by a mouldy diet containing ochratoxin A and penicillic acid, Veterinary Research Communications, 2001, 25, 3, 205-223
-Stoev, S. D., M. Paskalev, S. MacDonald, P.G. Mantle, Experimental one year ochratoxin A toxicosis in pigs, Experimental and Toxicologic Pathology, 2002, 53, 481-487
-Page 8, line 317-318 – The high contamination levels of OTA in 75 serum samples and 15 urine samples from extensive studies in 5 slaughterhouses in Bulgaria with mean levels in serum being higher in spring (60,88 µg/kg) than in autumn (4,8-21,94 µg/kg) should be added in Table 3 (Stoev et al, 1998).
-Stoev, S. D., J. Stoeva, G. Anguelov, B. Hald, E. E. Creppy, B. Radic, Haematological, biochemical and toxicological investigations in spontaneous cases with different frequency of porcine nephropathy in Bulgaria, Journal of Veterinary Medicine, Series A, 1998c, 45, 229-236
-Page 8, line 332 – The presence of OTA in 82 kidney samples from extensive studies in 5 slaughterhouses in Bulgaria with mean levels ranging between 1,5-7,17 µg/kg (ppm) (around 80-100% positive) should be added in Table 3 (Stoev et al, 1998).
-Stoev, S. D., B. Hald and P. Mantle, Porcine nephropathy in Bulgaria: a progressive syndrome of complex of uncertain (mycotoxin) etiology, The Veterinary Record, 1998, 142, 190-194.
-There are some contradictions in the “Conclusions” of this paper, which should be avoided – for example in the last paragraph of the chapter “9. Risk assessment of the human exposure to OTA by the consumption of pork meat and derived products” the author state that “Human exposure to OTA seems to be associated predominantly with the consumption of contaminated plant-derived products, and only to a minor extent to foods of animal origin…” and, thereafter, in the second paragraph of the “Conclusions” there is just the oposit statement that “Pork meat and meat products are considered as important sources to the chronic dietary exposure to OTA in humans.”
-The possible preventive measures to decrease human exposure from pork meat or offal from slaughtered pigs could be suggested in the Conclusion or in the Chapter “9 Risk assessment” of this paper such as changing the feed source for pigs before the slaughter time or condemnation of some of the offals, etc., in order to increase its usefulness for readers.
Author Response
Response to the reviewers’ comments on the manuscript entitled
“Ochratoxin A in slaughtered pigs and pork products”
(Manuscript ID Toxins-1498025)
Response to Review Report 3
The authors would like to thank the reviewer for his valuable comments that helped us to improve the quality of our manuscript.
The manuscript was revised according to reviewer’s 3 comments. The revised text was marked in light blue color.
Reviewer 3
This review paper suffers from unsufficient study in the available literature in regard to OTA contamination in pork meat, kidney or blood in various European countries and from some inaccuracies in the quoted data, and therefore a major revision should be done.
Response: Your comment was adopted; the text was checked and revised accordingly. Please see response to specific comments.
Part of information given in some chapters such as “3. Physicochemical properties of OTA” or “6. Methods for the detection and determination of OTA” is well known and not in line of the main focus of this paper – see the specific comments. Such information is not quite helpful for clarifying the way of OTA contamination of cereal and cereal by products, which are important ingredients in pig feeds and are responsible for OTA contamination in pork meat and offal – see the specific comments. The focus in such chapters should be only on the suggested methods and procedures for the detection of OTA in pork meat by-products.
The possible preventive measures to decrease human exposure from pork meat or offal from slaughtered pigs should be suggested in the Conclusion or in the Chapter 9 Risk assessment in this paper in order to increase its usefulness for readers.
Also, some of the given information in this paper is not quite correct and even wrong, e.g. the statement that Bulgarian mycotoxic porcine nephropathy is different from that described in South Africa, because just the oposit is true – it is very similar to it and both nephropathies Bulgarian and South African have multi-mycotoxic etiology in contrast to the classical porcine nephropathy as described in Denmark.
In addition, there are many inaccuracies in the thoughts expression or wrong using of unappropriate terms such as “OTA nfection of proximal tubules of kidneys” or “OTA infected pigs”, etc. It is well known that OTA is not an infectious agent, but mycotoxin, and therefore, it cannot infect the proximal tubules or the pigs.
There are also some contradictions in the “Conclusions” of this paper, where the authors state that “Human exposure to OTA seems to be associated predominantly with the consumption of contaminated plant-derived products, and only to a minor extent to foods of animal origin…” and, thereafter, in the second paragraph of the “Conclusions” there is just the oposit statement that “Pork meat and meat products are considered as important sources to the chronic dietary exposure to OTA in humans.”
Response: Your comments were adopted, and the text revised accordingly. Please see response to specific comments.
Specific comments:
-Page 1, line 4 and lines 26-27 – It should added, that OTA was also found to possess carcinogenic effect in animals and poultry, in addition to its nephrotoxic, hepatotoxic, teratogenic, neurotoxic, genotoxic and immunotoxic effects in animals [see Stoev, 2015, 2020, 2021] in the sentence “Several studies have indicated the nephrotoxic, immunotoxic, teratogenic, embryotoxic, genotoxic AND CARCINOGENIC properties of OTA….”
-Stoev, S. D., Studies on carcinogenic and toxic effects of ochratoxin A in chicks, Special issue “Ochratoxins”, Toxins, 2010a, 2, 649-664
-Stoev, S.D., Long term preliminary studies on toxic and carcinogenic effect of individual or simultaneous exposure to ochratoxin A and penicillic acid in mice, Toxicon, 2020, 184, 192–201
-Stoev, S.D., Follow up long term preliminary studies on carcinogenic and toxic effects of ochratoxin A in rats and the putative protection of phenylalanine, Toxicon, 2021, 190, 41-49
Response: Your comments were adopted, and the text revised accordingly (lines 5, 26-28 of revised version of the manuscript).
-Page 1, line 31-32 – in the sentence “OTA found in consumed feedstuffs, can adversely affect the animal health, increase susceptibility to microbial infections…” it should be added that the same mycotoxin could also increase susceptibility to secondary bacterial infections in growing pigs and immunosuppression is the first expressed toxic effect of ochratoxin A (Stoev et al, 2000).
Response: Your comments were adopted, and the text revised accordingly (lines 32-35 of revised version of the manuscript).
-Stoev, S. D., D. Goundasheva, T. Mirtcheva, P. G. Mantle, Susceptibility to secondary bacterial infections in growing pigs as an early response in ochratoxicosis, Experimental and Toxicologic Pathology, 2000, 52, 287-296
-Page 2, line 79-82 The sentence “OTA toxigenic A. carbonarius has been usually identified in grapes or wine [34,65-79 69], coffee beans and cocoa [34,70-74], maize, peanuts, paprika [75,76] and dried chilli 80 [58]. A. niger are isolated from several foods such as raisins, grapes, maize, peanuts and 81 coffee [73,77-80]” is not in line of the focus of this paper to clarify the way of OTA contaminated of cereal and cereal by products, which are important ingredients in pig feeds and are responsible for OTA contamination in pork meat and offal.
Response: Your comments were adopted, and the text revised accordingly (lines 67-72, 79-83 of original version of the manuscript were removed, lines 68-72 of revised version of the manuscript).
-Page 2, line 85-87 – In addition to the given Penicillia species of “…P. chrysogenum, P. glycyrrhizacola, P. polonicum [84], P. brevicompactum, P. crustosum, P. olsonii, P. oxalicum 86 [85], P. nalgiovense, P. solitum and P. salamii [86]…..”, the strain P. commune was also reported to produce OTA [157].
Response: Your comments were adopted, and the text revised accordingly (line 77 of revised version of the manuscript).
-Page 4, line 145-160 – The physicochemical properties and the biotransformation of OTA in different animals described in the chapter “3. Physicochemical properties of OTA” are well known and it is not necessary to describe them again and again.
Response: The presentation of the physicochemical properties in the present form was asked by previous reviewers. According to your comment, the text was revised and some of the information of the original version of manuscript was removed (lines 151-159 of original version were deleted).
Page 5, line 165-167 – In the sentence “Innovative food-processing methods such as irradiation, cold plasma (CP), high‐pressure processing (HPP) may be an effective means of OTA decrease in foods [139,140]” it should be clarified that gamma or electron beam irradiation is mainly used to control the growth of ochratoxinA-producing fungi, but not to destroy OTA itself.
Response: Your comment was adopted, and the text was revised accordingly (lines 145-146 of revised version of the manuscript).
-Page 5, line 173-175 – In the sentence “……and can cause tumors in organs like kidneys and liver, both in animals and humans [1,2,3,31,142]…” some other organs and tissues should be added such as intestine, ureters, lung, oculi and muscles [see the papers above Stoev, 2015, 2020, 2021].
Response: Your comment was adopted, and the text was revised accordingly (lines 154-156 of revised version of the manuscript).
-Page 5, line 175-178 - In the sentence “Consumption of OTA contaminated feed can adversely affect the animals’ health and has been associated with several animal diseases, including, porcine nephropathy, avian ochratoxicosis [143,144] and carcinogenicity in rodents and poultry [145]” the previous 3 publications should be added here – see above.
Response: Your comment was adopted, and the text was revised accordingly (lines 156-159 of revised version of the manuscript).
-Page 5, line 193 and 195 – The sentence “…..OTA is usually causing kidney disease by infecting proximal tubules…” and the sentence “….observed frequently during the carcass inspection of infected pigs….” are not quite correct, because OTA is not an infectious agent, but mycotoxin, and therefore, it cannot infect the proximal tubules or pigs. These sentences should be corrected to: “…..OTA is usually causing kidney disease by DAMAGING proximal tubules…..”. and “….observed frequently during the carcass inspection of pigs WITH NEPHROPATHY PROBLEMS”….”
Response: Your comment was adopted, and the text was revised accordingly (lines 174-176 of revised version of the manuscript).
-Page 5, line 195-198 The sentence “It is differentiated morphologically from the classical mycotoxic porcine nephropathy described in Denmark and South Africa, both of multi-mycotoxic etiology, involving several mycotoxins including OTA as well as penicillic acid [158]” is not quite correct, because Bulgarian nephropathy is not different from this one described in South Africa, but just the oposit – it is very similar to it and both nephropathies Bulgarian and South African have multi-mycotoxic etiology in contrast to the classical porcine nephropathy as described in Denmark.
Response: Your comment was adopted, and the text was revised accordingly (lines 176-179 of revised version of the manuscript).
-Page 6, line 216 – In this paragraph it should be explained that in mice, approximately 33% of OTA is eliminated through the “hepatobiliary route of excretion and enterohepatic recirculation of OTA” in mice and rats is mainly responsible for the liver damages in these species [131]. The quotation of reference “[137]” is not appropriate and adequate to be given here.
Response: Your comment was adopted, and the text was revised accordingly (lines 198-201 of revised version of the manuscript).
-Page 6, line 233 – The maximum permited content of mycotoxins in human food and animal feed are given not only in EC Regulation No 1881/2006, but also in EC Directive 2002/32/EC, and EC Recommendations 2006/576/EC and 2013/165/EU) (see the review paper Stoev, 2015 ).
-Stoev, S. D. Foodborne mycotoxicoses, risk assessment and underestimated hazard of masked mycotoxins and joint mycotoxin effects or interaction, Environmental Toxicology and Pharmacology, 2015, 9, 794–809.
Response: In these lines of the text the regulation on OTA in foods, not in animal feeds, are presented. In EC Directive 2002/32/EC, and EC Recommendations 2013/165/EU no regulation limits for OTA are included. EC Recommendations 2006/576/EC are presented in lines 105-107 of original version of the manuscript (lines 95-97 of revised version of manuscript) where the regulation on OTA in animal feeds are presented. The reference was added (line 97).
-Page 6-7, lines 246-303 – The most of information given in chapter “6. Methods for the detection and determination of OTA” is well known and not in line of the main focus of this paper, and therefore, it should be omitted or the focus should be only on the suggested methods and procedures for the detection of OTA in pork meat by-products.
Response: Your comment was adopted, and the text was revised accordingly (lines 248-257 of original version were deleted).
-Page 7, line 305-308 The studed relationships between the levels of OTA in fed forages and blood of pigs (Stoev et al, 2001) could be mentioned in Table 2, in addition to the possible relationships between the levels of OTA in blood and the continuance of OTA exposure or the age of the pigs (Stoev et al, 2002).
-Stoev, S.D., Vitanov, S., Anguelov, G., Petkova-Bocharova, T., Creppy, E. E. Experimental mycotoxic nephropathy in pigs provoked by a mouldy diet containing ochratoxin A and penicillic acid, Veterinary Research Communications, 2001, 25, 3, 205-223
-Stoev, S. D., M. Paskalev, S. MacDonald, P.G. Mantle, Experimental one year ochratoxin A toxicosis in pigs, Experimental and Toxicologic Pathology, 2002, 53, 481-487
Response: Your comment was adopted, and the text was revised accordingly (lines 284-285 of revised version of the manuscript & reference 162).
-Page 8, line 317-318 – The high contamination levels of OTA in 75 serum samples and 15 urine samples from extensive studies in 5 slaughterhouses in Bulgaria with mean levels in serum being higher in spring (60,88 µg/kg) than in autumn (4,8-21,94 µg/kg) should be added in Table 3 (Stoev et al, 1998).
-Stoev, S. D., J. Stoeva, G. Anguelov, B. Hald, E. E. Creppy, B. Radic, Haematological, biochemical and toxicological investigations in spontaneous cases with different frequency of porcine nephropathy in Bulgaria, Journal of Veterinary Medicine, Series A, 1998c, 45, 229-236
-Page 8, line 332 – The presence of OTA in 82 kidney samples from extensive studies in 5 slaughterhouses in Bulgaria with mean levels ranging between 1,5-7,17 µg/kg (ppm) (around 80-100% positive) should be added in Table 3 (Stoev et al, 1998).
-Stoev, S. D., B. Hald and P. Mantle, Porcine nephropathy in Bulgaria: a progressive syndrome of complex of uncertain (mycotoxin) etiology, The Veterinary Record, 1998, 142, 190-194.
Response: Your comments were adopted, the reference 176, 177 was added and the table was revised accordingly.
-There are some contradictions in the “Conclusions” of this paper, which should be avoided – for example in the last paragraph of the chapter “9. Risk assessment of the human exposure to OTA by the consumption of pork meat and derived products” the author state that “Human exposure to OTA seems to be associated predominantly with the consumption of contaminated plant-derived products, and only to a minor extent to foods of animal origin…” and, thereafter, in the second paragraph of the “Conclusions” there is just the oposit statement that “Pork meat and meat products are considered as important sources to the chronic dietary exposure to OTA in humans.”
Response: Your comment was adopted, and the text was revised accordingly.
-The possible preventive measures to decrease human exposure from pork meat or offal from slaughtered pigs could be suggested in the Conclusion or in the Chapter “9 Risk assessment” of this paper such as changing the feed source for pigs before the slaughter time or condemnation of some of the offals, etc., in order to increase its usefulness for readers.
Response: Your comment was adopted, and the text was revised accordingly.

Reviewer 4 Report
Dear
It is quite difficult to read this review. Some recent papers are missing (e.g. "Determination of ochratoxin A in edible pork offal: intra-laboratory validation study and estimation of the daily intake via kidney consumption in Belgium. Mycotoxin Research 2021; DOI: 10.1007/s12550-020-00415-7."
In addition, authors should discuss about the fact that among farmed animals, pigs are particularly sensitive to OTA. In experimental conditions, pigs fed with natural OTA contaminated feed showed about three to four times higher levels of OTA than pigs fed with crystalline OTA contaminated feed at the same concentration level.
They should also discuss about the EFSA document in risk assessment of OTA in food. Indeed, tolerable weekly intake is no longer valid because of carcinogenicity and, instead, a margin of exposure applies.
Please make short Tables 2, 3 and 4 in this review.
Please discuss also "while contamination of feed and food takes place during the field period or storage of raw materials for feed and food, cured meat can become contaminated during processing".
Best regards
Author Response
Response to the reviewers’ comments on the manuscript entitled
“Ochratoxin A in slaughtered pigs and pork products”
(Manuscript ID Toxins-1498025)
Response to Review Report 4
The authors would like to thank the reviewer for his valuable comments that helped us to improve the quality of our manuscript.
The manuscript was revised according to reviewer’s 2 comments. The revised text was marked in green color.
Reviewer 4
Dear
It is quite difficult to read this review. Some recent papers are missing (e.g. "Determination of ochratoxin A in edible pork offal: intra-laboratory validation study and estimation of the daily intake via kidney consumption in Belgium. Mycotoxin Research 2021; DOI: 10.1007/s12550-020-00415-7."
Response: Your comment was adopted, the text and the Table were revised accordingly (lines 310-311, 350-351 of revised version of the manuscript).. (Please note that this work haw been published after our submission).
In addition, authors should discuss about the fact that among farmed animals, pigs are particularly sensitive to OTA. In experimental conditions, pigs fed with natural OTA contaminated feed showed about three to four times higher levels of OTA than pigs fed with crystalline OTA contaminated feed at the same concentration level.
Response: The information you asked are found in our lines text (41,42, 173-174 of revised version of manuscript). Your comment was adopted, and the text was revised accordingly (lines 285-287 of revised version of the manuscript).
They should also discuss about the EFSA document in risk assessment of OTA in food. Indeed, tolerable weekly intake is no longer valid because of carcinogenicity and, instead, a margin of exposure applies.
Response: The EFSA scientific opinion is discussed in section 9.
Please make short Tables 2, 3 and 4 in this review.
Response: The presentation of data in Tables 2,3 and 4 was asked in the form by other reviewers.
Please discuss also "while contamination of feed and food takes place during the field period or storage of raw materials for feed and food, cured meat can become contaminated during processing".
Response: Your comment was adopted, and the text was revised accordingly (lines 25-26 of revised version of the manuscript). Please also see lines 47-51 of revised version of manuscript.
Best regards

Round 2
Reviewer 3 Report
Part of information given in some chapters such as “3. Physicochemical properties of OTA” or “6. Methods for the detection and determination of OTA”, which was not in line of the main focus of this paper is now shortened. Such information was not quite helpful for clarifying the way of OTA contamination of cereal and cereal by products used for pig feed. The focus of the above mentioned chapters is now mainly on the suggested methods and procedures for the detection of OTA in pork meat by-products, which is the subject of this review paper.
All mistakes or wrong usage of some terms/words are now corrected/avoided accordingly.
The contradictions seen in the Conclusions are now avoided accordingly.
Some minor mistakes are only observed as follow:
-Page 1, line 40 – a mistake is seen in the spelling of the word „comtributing” – it should be “contributing”
-Numbering of the pages is wrong and mixed
-The sentence in the second paragraph of Conclusion “Pork meat and meat products are considered as important sources to the chronic dietary exposure to OTA in humans, among foods of animal origin.” could be improved by changing the place of its first and last part „Among foods of animal origin, pork meat and meat products are considered as important sources to the chronic dietary exposure to OTA in humans.“
Author Response
Response to the reviewers’ comments on the manuscript entitled
“Ochratoxin A in slaughtered pigs and pork products”
(Manuscript ID Toxins-1498025)
2nd Round
Response to the Review Report 3
The authors would like to thank the reviewer for his valuable comments that helped us to improve the quality of our manuscript.
The manuscript was revised according to reviewer’s 3 comments. The revised text was marked in yellow color.
Reviewer 3
Part of information given in some chapters such as “3. Physicochemical properties of OTA” or “6. Methods for the detection and determination of OTA”, which was not in line of the main focus of this paper is now shortened. Such information was not quite helpful for clarifying the way of OTA contamination of cereal and cereal by products used for pig feed. The focus of the above mentioned chapters is now mainly on the suggested methods and procedures for the detection of OTA in pork meat by-products, which is the subject of this review paper.
All mistakes or wrong usage of some terms/words are now corrected/avoided accordingly.
The contradictions seen in the Conclusions are now avoided accordingly.
Response: Thank you for your comments.
Some minor mistakes are only observed as follow:
-Page 1, line 40 – a mistake is seen in the spelling of the word „comtributing” – it should be “contributing”
Response: Your comment was adopted (line 45 of revised version of the manuscript).
-Numbering of the pages is wrong and mixed
Response: Your comment was adopted, and the numbering of the pages was corrected.
-The sentence in the second paragraph of Conclusion “Pork meat and meat products are considered as important sources to the chronic dietary exposure to OTA in humans, among foods of animal origin.” could be improved by changing the place of its first and last part „Among foods of animal origin, pork meat and meat products are considered as important sources to the chronic dietary exposure to OTA in humans.“
Response: Your comment was adopted, and the text was revised accordingly (Lines 440-442 of revised version of the manuscript).

Author Response
Response to the reviewers’ comments on the manuscript entitled
“Ochratoxin A in slaughtered pigs and pork products”
(Manuscript ID Toxins-1498025)
2nd Round
Response to the Review Report 4
The authors would like to thank the reviewer for his review and valuable comments that helped us to improve the quality of our manuscript.
